# Multiscale Fields of Patterns

**Pedro F. Felzenszwalb**
Brown University
Providence, RI 02906
`pff@brown.edu`

**John G. Oberlin**
Brown University
Providence, RI 02906
`john_oberlin@brown.edu`

## Abstract

We describe a framework for defining high-order image models that can be used in a variety of applications. The approach involves modeling local patterns in a multiscale representation of an image. Local properties of a coarsened image reflect non-local properties of the original image. In the case of binary images local properties are defined by the binary patterns observed over small neighborhoods around each pixel. With the multiscale representation we capture the frequency of patterns observed at different scales of resolution. This framework leads to expressive priors that depend on a relatively small number of parameters. For inference and learning we use an MCMC method for block sampling with very large blocks. We evaluate the approach with two example applications. One involves contour detection. The other involves binary segmentation.

## 1 Introduction

Markov random fields are widely used as priors for solving a variety of vision problems such as image restoration and stereo [5, 8]. Most of the work in the area has concentrated on low-order models involving pairs of neighboring pixels. However, it is clear that realistic image priors need to capture higher-order properties of images.

In this paper we describe a general framework for defining high-order image models that can be used in a variety of applications. The approach involves modeling local properties in a multiscale representation of an image. This leads to a natural low-dimensional representation of a high-order model. We concentrate on the problem of estimating binary images. In this case local image properties can be captured by the binary patterns in small neighborhoods around each pixel.

We define a *Field of Patterns* (FoP) model using an energy function that assigns a cost to each 3x3 pattern observed in an image pyramid. The cost of a pattern depends on the scale where it appears. Figure 1 shows a binary image corresponding to a contour map from the Berkeley segmentation dataset (BSD) [12, 2] and a pyramid representation obtained by repeated coarsening. The 3x3 patterns we observe after repeated coarsening depend on large neighborhoods of the original image. These coarse 3x3 patterns capture non-local image properties. We train models using a maximum-likelihood criteria. This involves selecting pattern costs making the expected frequency of patterns in a random sample from the model match the average frequency of patterns in the training images. Using the pyramid representation the model matches frequencies of patterns at each resolution.

In practice we use MCMC methods for inference and learning. In Section 3 we describe an MCMC sampling algorithm that can update a very large area of an image (a horizontal or vertical band of pixels) in a single step, by combining the forward-backward algorithm for one-dimensional Markov models with a Metropolis-Hastings procedure.

We evaluated our models and algorithms on two different applications. One involves contour detection. The other involves binary segmentation. These two applications require very different image priors. For contour detection the prior should encourage a network of thin contours, while for bi-

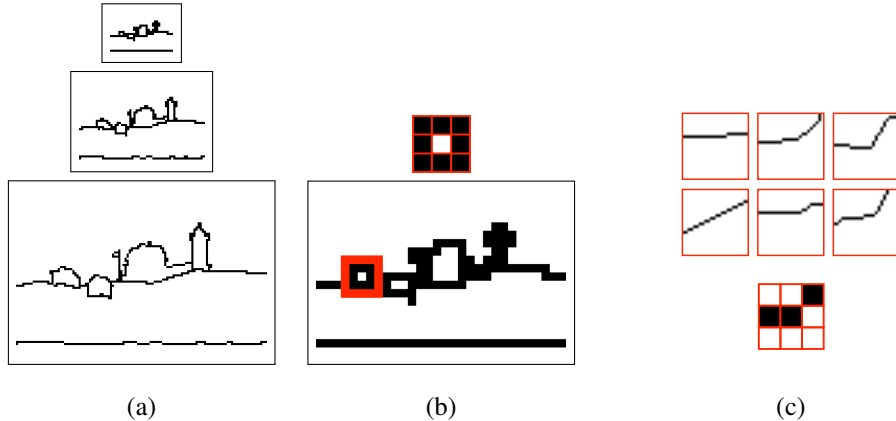

<center>(a)                  (b)                  (c)</center>

Figure 1: (a) Multiscale/pyramid representation of a contour map. (b) Coarsest image scaled up for better visualization, with a 3x3 pattern highlighted. The leftmost object in the original image appears as a 3x3 "circle" pattern in the coarse image. (c) Patches of contour maps (top) that coarsen to a particular 3x3 pattern (bottom) after reducing their resolution by a factor of 8.

nary segmentation the prior should encourage spatially coherent masks. In both cases we can design effective models using maximum-likelihood estimation.

## 1.1 Related Work

FRAME models [24] and more recently Fields of Experts (FoE) [15] defined high-order energy models using the response of linear filters. FoP models are closely related. The detection of 3x3 patterns at different resolutions corresponds to using *non-linear* filters of increasing size. In FoP we have a fixed set of pre-defined non-linear filters that detect common patterns at different resolutions. This avoids filter learning, which leads to a non-convex optimization problem in FoE.

A restricted set of 3x3 binary patterns was considered in [6] to define priors for image restoration. Binary patterns were also used in [17] to model curvature of a binary shape. There has been recent work on inference algorithms for CRFs defined by binary patterns [19] and it may be possible to develop efficient inference algorithms for FoP models using those techniques.

The work in [23] defined a variety of multiresolution models for images based on a quad-tree representation. The quad-tree leads to models that support efficient learning and inference via dynamic programming, but such models also suffer from artifacts due to the underlying tree-structure. The work in [7] defined binary image priors using deep Boltzmann machines. Those models are based on a hierarchy of hidden variables that is related to our multiscale representation. However in our case the multiscale representation is a deterministic function of the image and does not involve extra hidden variables as [7]. The approach we take to define a multiscale model is similar to [9] where local properties of subsampled signals where used to model curves.

One of our motivating applications involves detecting contours in noisy images. This problem has a long history in computer vision, going back at least to [16], who used a type of Markov model for detecting salient contours. Related approaches include the stochastic completion field in [22, 21], spectral methods [11], the curve indicator random field [3], and the more recent work in [1].

## 2 Fields of Patterns (FoP)

Let $\mathcal{G} = [n] \times [m]$ be the grid of pixels in an $n$ by $m$ image. Let $x = \{x(i,j) \mid (i,j) \in \mathcal{G}\}$ be a hidden binary image and $y = \{y(i,j) \mid (i,j) \in \mathcal{G}\}$ be a set of observations (such as a grayscale or color image). Our goal is to estimate $x$ from $y$.

We define $p(x|y)$ using an energy function that is a sum of two terms,

$$p(x|y) = \frac{1}{Z(y)} \exp(-E(x,y)) \quad E(x,y) = E_{\text{FoP}}(x) + E_{\text{data}}(x,y) \tag{1}$$

It is sometimes useful to think of $E_{\text{FoP}}(x)$ as a model for binary images and $E_{\text{data}}(x, y)$ as a data model even though technically there is no such distinction in a conditional model.

## 2.1 Singlescale FoP Model

The singlescale FoP model is one of the simplest energy models that can capture the basic properties of contour maps or other images that contain thin objects. We use $x[i, j]$ to denote the binary pattern defined by $x$ in the 3x3 window centered at pixel $(i, j)$, treating values outside of the image as 0. A singlescale FoP model is defined by the local patterns in $x$,

$$E_{\text{FoP}}(x) = \sum_{(i,j) \in \mathcal{G}} V(x[i,j]). \tag{2}$$

Here $V$ is a potential function assigning costs (or energies) to binary patterns. Note that there are 512 possible binary patterns in a 3x3 window. We can make the model invariant to rotations and mirror symmetries by tying parameters together. The resulting model has 102 parameters (some patterns have more symmetries than others) and can be learned from smaller datasets. We used invariant models for all of the experiments reported in this paper.

## 2.2 Multiscale FoP Model

To capture non-local statistics we look at local patterns in a multiscale representation of $x$. For a model with $K$ scales let $\sigma(x) = x^0, \ldots, x^{K-1}$ be an image pyramid where $x^0 = x$ and $x^{k+1}$ is a coarsening of $x^k$. Here $x^k$ is a binary image defined over a grid $\mathcal{G}^k = [n/2^k] \times [m/2^k]$. The coarsening we use in practice is defined by a logical OR operation,

$$x^{k+1}(i, j) = x^k(2i, 2j) \vee x^k(2i + 1, 2j) \vee x^k(2i, 2j + 1)^k \vee x^k(2i + 1, 2j + 1) \tag{3}$$

This particular coarsening maps connected objects at one scale of resolution to connected objects at the next scale, but other coarsenings may be appropriate in different applications.

A multiscale FoP model is defined by the local patterns in $\sigma(x)$,

$$E_{\text{FoP}}(x) = \sum_{k=0}^{K-1} \sum_{(i,j) \in \mathcal{G}^k} V^k(x^k[i, j]). \tag{4}$$

This model is parameterized by $K$ potential functions $V^k$., one for each scale in the pyramid $\sigma(x)$. In many applications we expect the frequencies of a 3x3 pattern to be different at each scale. The potential functions can encourage or discourage specific patterns to occur at specific scales.

Note that $\sigma(x)$ is a deterministic function and the pyramid representation does not introduce new random variables. The pyramid simply defines a convenient way to specify potential functions over large regions of $x$. A single potential function in a multiscale model can depend on a large area of $x$ due to the coarsenings. For large enough $K$ (proportional to $\log$ of the image size) the Markov blanket of a pixel can be the whole image.

While the experiments in Section 5 use the conditional modeling approach specified by Equation (1), we can also use $E_{\text{FoP}}$ to define priors over binary images. Samples from these priors illustrate the information that is captured by a FoP model, specially the added benefit of the multiscale representation. Figure 2 shows samples from FoP priors trained on contour maps of natural images.

The empirical studies in [14] suggest that low-order Markov models can not capture the empirical length distribution of contours in natural images. A multiscale FoP model can control the size distribution of objects much better than a low-order MRF. After coarsening the diameter of an object goes down by a factor of approximately two, and eventually the object is mapped to a single pixel. The scale at which this happens can be captured by a 3x3 pattern with an "on" pixel surrounded by "off" pixels (this assumes there are no other objects nearby). Since the cost of a pattern depends on the scale at which it appears we can assign a cost to an object that is based loosely upon its size.

## 2.3 Data Model

Let $y$ be an input image and $\sigma(y)$ be an image pyramid computed from $y$. Our data models are defined by sums over pixels in the two pyramids $\sigma(x)$ and $\sigma(y)$. In our experiments $y$ is a graylevel

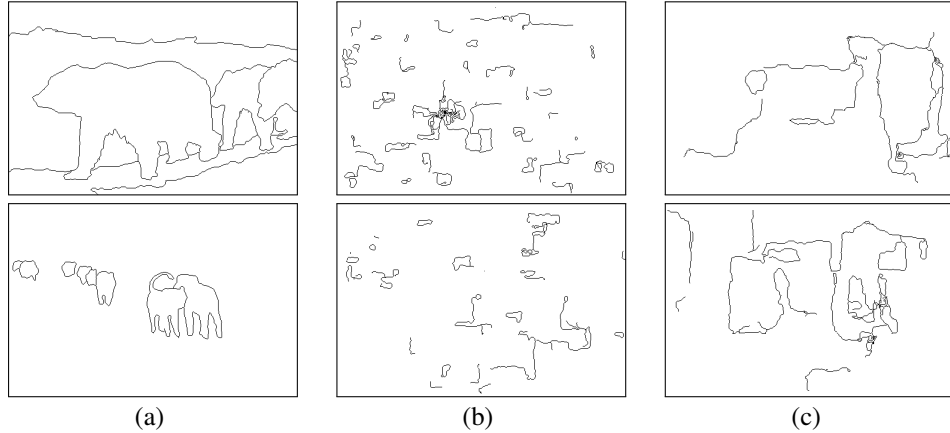

|(a)|(b)|(c)|

Figure 2: (a) Examples of training images $T$ extracted from the BSD. (b) Samples from a singlescale FoP prior trained on $T$. (c) Samples from a multiscale FoP prior trained on $T$. The multiscale model is better at capturing the lengths of contours and relationships between them.

image with values in $\{0, \ldots, M-1\}$. The pyramid $\sigma(y)$ is defined in analogy to $\sigma(x)$ except that we use a local average for coarsening instead of the logical OR,

$$y^{k+1}(i,j) = \lfloor (y^k(2i,2j) + y^k(2i+1,2j) + y^k(2i,2j+1) + y^k(2i+1,2j+1))/4 \rfloor \quad (5)$$

The data model is parameterized by $K$ vectors $D^0, \ldots, D^{K-1} \in \mathbb{R}^M$

$$E_{\text{data}}(x,y) = \sum_{k=0}^{K-1} \sum_{(i,j) \in \mathcal{G}^k} x^k(i,j) D^k(y^k(i,j)) \quad (6)$$

Here $D^k(y^k(i,j))$ is an observation cost incurred when $x^k(i,j) = 1$. There is no need to include an observation cost when $x^k(i,j) = 0$ because only energy differences affect the posterior $p(x|y)$.

We note that it would be interesting to consider data models that capture complex relationships between local patterns in $\sigma(x)$ and $\sigma(y)$. For example a local maximum in $y^k(i,j)$ might give evidence for $x^k(i,j) = 1$, or a particular 3x3 pattern in $x^k[i,j]$.

### 2.4 Log-Linear Representation

The energy function $E(x,y)$ of a FoP model can be expressed by a dot product between a vector of model parameters $w$ and a feature vector $\phi(x,y)$. The vector $\phi(x,y)$ has one block for each scale. In the $k$-th block we have: (1) 512 (or 102 for invariant models) entries counting the number of times each 3x3 pattern occurs in $x^k$; and (2) $M$ entries counting the number of times each possible value for $y(i,j)$ occurs where $x^k(i,j) = 1$. The vector $w$ specifies the cost for each pattern in each scale ($V^k$) and the parameters of the data model ($D^k$). We then have that $E(x,y) = w \cdot \phi(x,y)$. This log-linear form is useful for learning the model parameters as described in Section 4.

## 3 Inference with a Band Sampler

In inference we have a set of observations $y$ and want to estimate $x$. We use MCMC methods [13] to draw samples from $p(x|y)$ and estimate the posterior marginal probabilities $p(x(i,j) = 1|y)$. Sampling is also used for learning model parameters as described in Section 4.

In a block Gibbs sampler we repeatedly update $x$ by picking a block of pixels $B$ and sampling new values for $x_B$ from $p(x_B|y, x_{\overline{B}})$. If the blocks are selected appropriately this defines a Markov chain with stationary distribution $p(x|y)$.

We can implement a block Gibbs sampler for a multiscale FoP model by keeping track of the image pyramid $\sigma(x)$ as we update $x$. To sample from $p(x_B|y, x_{\overline{B}})$ we consider each possible configuration

for $x_B$. We can efficiently update $\sigma(x)$ to reflect a possible configuration for $x_B$ and evaluate the terms in $E(x, y)$ that depend on $x_B$. This takes $O(K|B|)$ time for each configuration for $x_B$. This in turn leads to an $O(K|B|2^{|B|})$ time algorithm for sampling from $p(x_B|y, x_{\bar{B}})$. The running time can be reduced to $O(K2^{|B|})$ using Gray codes to iterate over configurations for $x_B$.

Here we define a *band sampler* that updates *all* pixels in a horizontal or vertical band of $x$ in a single step. Consider an $n$ by $m$ image $x$ and let $B$ be a horizontal band of pixels with $h$ rows. Since $|B| = mh$ a straightforward implementation of block sampling for $B$ is completely impractical. However, for an Ising model we can generate samples from $p(x_B|y, x_{\bar{B}})$ in $O(m2^{2h})$ time using the forward-backward algorithm for Markov models. We simply treat each column of $B$ as a single variable with $2^h$ possible states. A similar idea can be used for FoP models.

Let $S$ be a state space where a state specifies a joint configuration of binary values for the pixels in a column of $B$. Note that $|S| = 2^h$. Let $z_1, \ldots, z_m$ be a representation of $x_B$ in terms of the state of each column. For a singlescale FoP model the distribution $p(z_1, \ldots, z_n|y, x_{\bar{B}})$ is a 2nd-order Markov model. This allows for efficient sampling using forward weights computed via dynamic programming. Such an algorithm takes $O(m2^{3h})$ time to generate a sample from $p(x_B|y, x_{\bar{B}})$, which is efficient for moderate values of $h$.

In a multiscale FoP model the 3x3 patterns in the upper levels of $\sigma(x)$ depend on many columns of $B$. This means $p(z_1, \ldots, z_n|x_{\bar{B}})$ is no longer 2nd-order. Therefore instead of sampling $x_B$ directly we use a Metropolis-Hastings approach. Let $p$ be a multiscale FoP model we would like to sample from. Let $q$ be a singlescale FoP model that approximates $p$. Let $x$ be the current state of the Markov chain and $x'$ be a proposal generated by the singlescale band sampler for $q$. We accept $x'$ with probability $\min(1, ((p(x'|y)q(x|y))/(p(x|y)q(x'|y))))$. Efficient computation of acceptance probabilities can be done using the pyramid representations of $x$ and $y$. For each proposal we update $\sigma(x)$ to $\sigma(x')$ and compute the difference in energy due to the change under both $p$ and $q$.

One problem with the Metropolis-Hastings approach is that if proposals are rejected very often the resulting Markov chain mixes slowly. We can avoid this problem by noting that most of the work required to generate a sample from the proposal distribution involves computing forward weights that can be re-used to generate other samples. Each step of our band sampler for a multiscale FoP model picks a band $B$ (horizontal or vertical) and generates many proposals for $x_B$, accepting each one with the appropriate acceptance probability. As long as one of the proposals is accepted the work done in computing forward weights is not wasted.

## 4   Learning

We can learn models using maximum-likelihood and stochastic gradient descent. This is similar to what was done in [24, 15, 20]. But in our case we have a conditional model so we maximize the conditional likelihood of the training examples.

Let $T = \{(x_1, y_i), \ldots, (x_N, y_N)\}$ be a training set with $N$ examples. We define the training objective using the negative log-likelihood of the data plus a regularization term. The regularization ensures no pattern is too costly. This helps the Markov chains used during learning and inference to mix reasonably fast. Let $L(x_i, y_i) = -\log p(x_i|y_i)$. The training objective is given by

$$O(w) = \frac{\lambda}{2}||w||^2 + \sum_{i=1}^{N} L(x_i, y_i). \tag{7}$$

This objective is convex and

$$\nabla O(w) = \lambda w + \sum_{i=1}^{N} \phi(x_i, y_i) - E_{p(x|y_i)}[\phi(x, y_i)]. \tag{8}$$

Here $E_{p(x|y_i)}[\phi(x, y_i)]$ is the expectation of $\phi(x, y_i)$ under the posterior $p(x|y_i)$ defined by the current model parameters $w$. A stochastic approximation to the gradient $\nabla O(w)$ can be obtained by sampling $x_i'$ from $p(x|y_i)$. Let $\eta$ be a learning rate. In each stochastic gradient descent step we sample $x_i'$ from $p(x|y_i)$ and update $w$ as follows

$$w := w - \eta(\lambda w + \sum_{i=1}^{N} \phi(x_i, y_i) - \phi(x_i', y_i)). \tag{9}$$

To sample the $x_i'$ we run $N$ Markov chains, one for each training example, using the band sampler from Section 3. After each model update we advance each Markov chain for a small number of steps using the latest model parameters to obtain new samples $x_i'$.

## 5 Applications

To evaluate the ability of FoP to adapt to different problems we consider two different applications. In both cases we estimate hidden binary images $x$ from grayscale input images $y$. We used ground truth binary images obtained from standard datasets and synthetic observations. For the experiments described here we generate $y$ by sampling a value $y(i, j)$ for each pixel independently from a normal distribution with standard deviation $\sigma_y$ and mean $\mu_0$ or $\mu_1$, depending on $x(i, j)$,

$$y(i, j) \sim \mathcal{N}(\mu_{x(i,j)}, \sigma_y^2). \tag{10}$$

We have also done experiments with more complex data models but the results we obtained were similar to the results described here.

### 5.1 Contour Detection

The BSD [12, 2] contains images of natural scenes and manual segmentations of the most salient objects in those images. We used one manual segmentation for each image in the BSD500. From each image we generated a contour map $x$ indicating the location of boundaryes between segments in the image. To generate the observations $y$ we used $\mu_0 = 150$, $\mu_1 = 100$ and $\sigma_y = 40$ in Equation (10). Our training and test sets each have 200 examples. We first trained a 1-scale FoP model. We then trained a 4-level FoP model using the 1-level model as a proposal distribution for the band sampler (see Section 3). Training each model took 2 days on a 20-core machine. During training and testing we used the band sampler with $h = 3$ rows. Inference involves estimating posterior marginal probabilities for each pixel by sampling from $p(x|y)$. Inference on each image took 20 minutes on an 8-core machine.

For comparison we implemented a baseline technique using linear filters. Following [10] we used the second derivative of an elongated Gaussian filter together with its Hilbert transform. The filters had an elongation factor of 4 and we experimented with different values for the base standard deviation $\sigma_b$ of the Gaussian. The sum of squared responses of both filters defines an oriented energy map. We evaluated the filters at 16 orientations and took the maximum response at each pixel. We performed non-maximum suppression along the dominant orientations to obtain a thin contour map.

Figure 3 illustrates our results on 3 examples from the test set. Results on more examples are available in the supplemental material. For the FoP models we show the posterior marginal probabilities $p(x(i, j) = 1|y)$. The darkness of a pixel is proportional to the marginal probability. The FoP models do a good job suppressing noise and localizing the contours. The multiscale FoP model in particular gives fairly clean results despite the highly noisy inputs. The baseline results at lower $\sigma_b$ values suffer from significant noise, detecting many spurious edges. The baseline at higher $\sigma_b$ values suppresses noise at the expense of having poor localization and missing high-curvature boundaries.

For a quantitative evaluation we compute precision-recall curves for the different models by thresholding the estimated contour maps at different values. Figure 4 shows the precision-recall curves. The average precision (AP) was found by calculating the area under the precision-recall curves. The 1-level FoP model AP was 0.73. The 4-level FoP model AP was 0.78. The best baseline AP was 0.18 obtained with $\sigma_b = 1$. We have also done experiments using lower observation noise levels $\sigma_y$. With low observation noise the 1-level and 4-level FoP results become similar and baseline results improve significantly approaching the FoP results.

### 5.2 Binary Segmentation

For this experiment we obtained binary images from the Swedish Leaf Dataset [18]. We focused on the class of Rowan leaves because they have complex shapes. Each image defines a segmentation mask $x$. To generate the observations $y$ we used $\mu_0 = 150$, $\mu_1 = 100$ and $\sigma_y = 100$ in Equation (10). We used a higher $\sigma_y$ compared to the previous experiment because the 2D nature of masks makes it possible to recover them under higher noise. We used 50 examples for training and 25

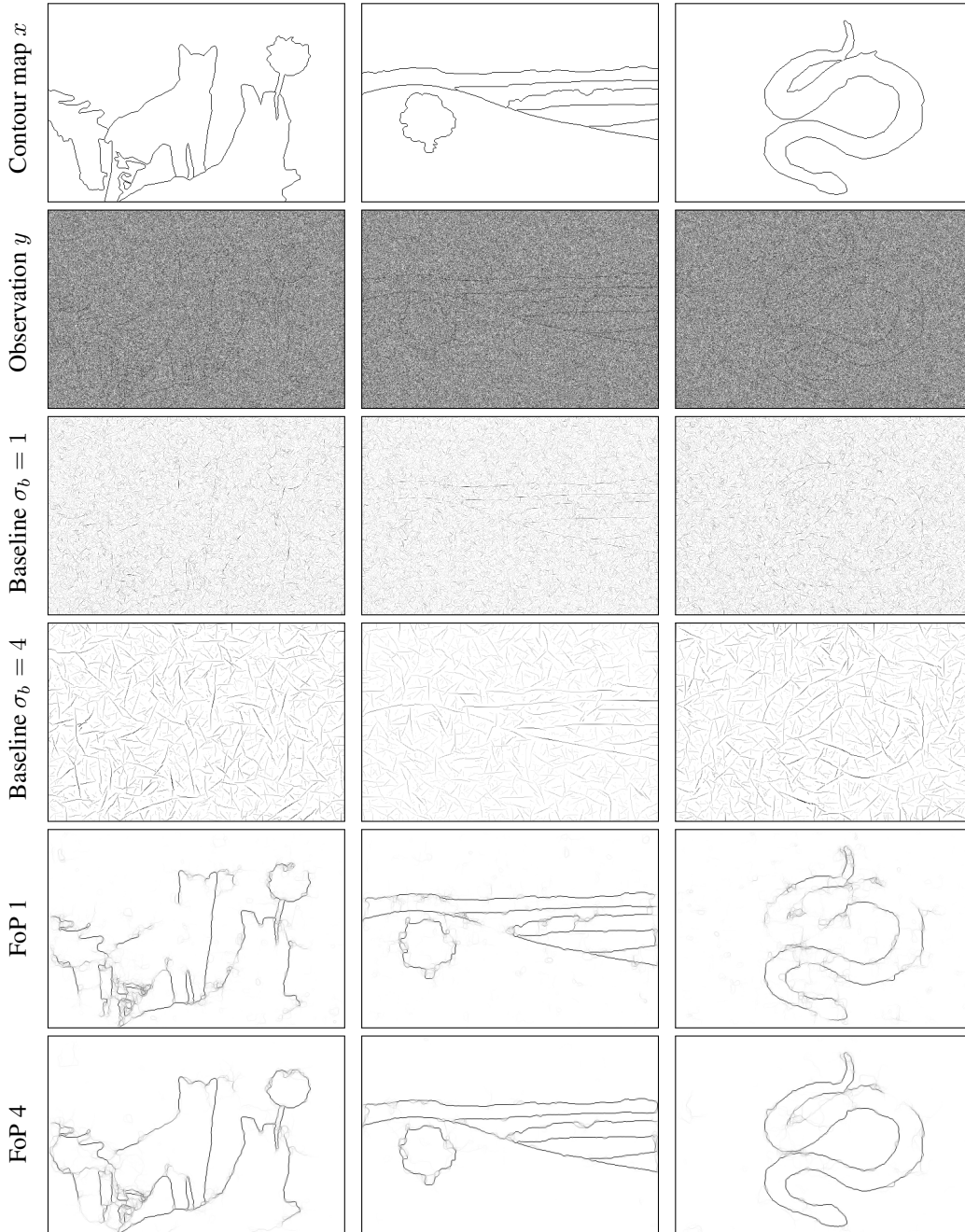

Figure 3: Contour detection results. Top-to-bottom: Hidden contour map $x$, input image $y$, output of oriented filter baseline with $\sigma_b = 1$ and $\sigma_b = 4$, output of 1-level and 4-level FoP model.

examples for testing. We trained FoP models with the same procedure and parameters used for the contour detection experiment. For a baseline, we used graph-cuts [5, 4] to perform MAP inference with an Ising model. We set the data term using our knowledge of the observation model and picked the pairwise discontinuity cost minimizing the per-pixel error rate in the test set.

Figure 5 illustrates the results of the different methods. Results on other images are available in the supplemental material. The precision-recall curves are in Figure 4. Graph-cuts yields a precision-recall point, with precision 0.893 and recall 0.916. The 1-level FoP model has a higher precision of 0.915 at the same recall. The 4-level FoP model raises the precision to 0.929 at the same recall. The

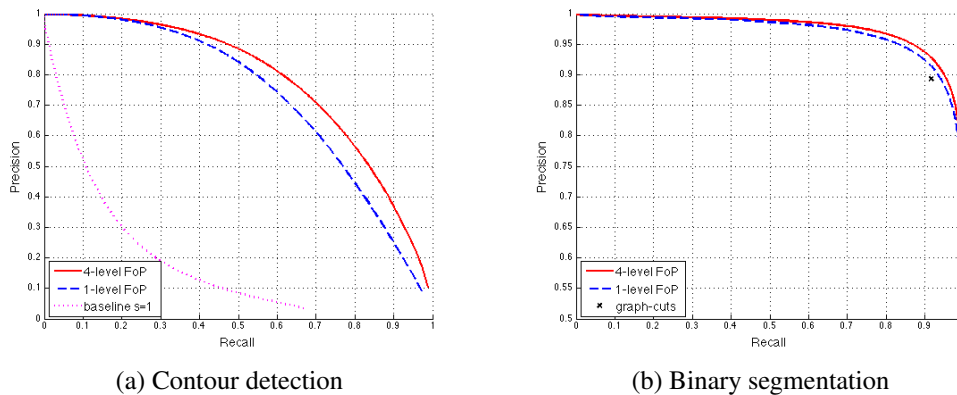

(a) Contour detection             (b) Binary segmentation

Figure 4: (a) Precision-recall curves for the contour detection experiment. (b) Precision-recall curves for the segmentation experiment (the graph-cuts baseline yields a single precision-recall point).

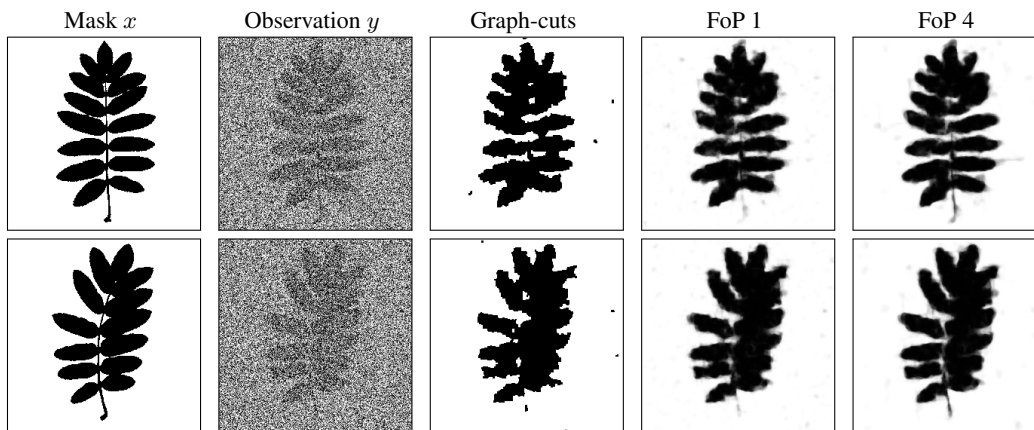

Figure 5: Binary segmentation examples. The 4-level FoP model does a better job recovering pixels near the object boundary and the stem of the leaves.

differences in precision are small because they are due to pixels near the object boundary but those are the hardest pixels to get right. There are clear differences that can be seen by visual inspection.

# 6 Conclusion

We described a general framework for defining high-order image models. The idea involves modeling local properties in a multiscale representation of an image. This leads to a natural low-dimensional parameterization for high-order models that exploits standard pyramid representations of images. Our experiments demonstrate the approach yields good results on two applications that require very different image priors, illustrating the broad applicability of our models. An interesting direction for future work is to consider FoP models for non-binary images.

# Acknowledgements

We would like to thank Alexandra Shapiro for helpful discussions and initial experiments related to this project. This material is based upon work supported by the National Science Foundation under Grant No. 1161282.

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
