[Reviews · NeurIPS 2014]

Submitted by Assigned_Reviewer_4

The paper proposes to extend the fields of experts model to a multiscale image representation, and using a combination of a binary latent variable model and a gray-level image model. The model is applied to images of contours embedded in noise, and black/white drawings embedded in noise, and the results show that the model can do a reasonable job at recovering the underlying binary process.

The overall idea here is useful and interesting - i.e., leveraging a multiscale model to capture long-range in addition to short-range structure in images. It is an approach long overdue in image models, which are often applied only to local patches at the highest-grain pixel level. So I think in that sense this paper would be of interest to NIPS as it points the way to a better model of images. However the particular application here seems rather contrived and artificial. The performance at detecting binary contours in noisy images is impressive, but at the same time it seems like a task invented for this particular algorithm. The binary mask example is somewhat more compelling, and the improvement over graph cuts is also good. But I am still left wondering what real world task you would actually use this particular model for.

Also it would be nice if the authors could show something about what the model has learned. Is there somewhat to visualize w and how it relates to the structure in images?

Summary: A good idea and good execution, but the examples used to demonstrate performance could be more compelling.

Submitted by Assigned_Reviewer_12

This paper presents a multiscale, generative model for binary patterns. The method works by building upon a multiscale representation for binary images defined by a restriction operator (along with an equivalent one for grayscale images). Each possible 3x3 pattern is assigned a different cost, An inference procedure using MCMC is proposed via a "band sampler" where entire rows or columns of the pyramid are sampled together.
The method is demonstrated to work on contour recovery from very noisy measurements with nice results.

This is an interesting paper with a compelling representation and inference framework.
I have some concerns regarding the clarity of presentation - I think this will require some work. A lot of the terms are not really explained, as well as each procedure - for example, in the learning part, it is not clear if the parameters of the model are learned from ground truth data, and if so, which one? which parameters are trained, and which are inferred from each image?

The experimental validation is interesting, though I would say a more "realistic" scenario would have been more convincing - like actually making use of this in a "real" boundary detection scenario. The samples sampled from the model are quite nice.

In terms of originality, this is good work - however, it misses a lot of the literature on binary patterns and natural images. Notably works by Gasper Tkacik and William Biallek - for example:

Thermodynamics of natural images, Greg J Stephens, Thierry Mora, Gasper Tkacik, William Bialek in arXiv

and there are several other which require at least some discussion.

In terms of significance this is an important subject and would be of interest to the community.

Summary: This is an interesting work with a compelling framework and interesting inference procedure. It does, however, leave something to be desired in clarity, experimental validation and related works.

Submitted by Assigned_Reviewer_32

The authors propose a way to define distributions over high-order image features, using a multiscale representation of binary image features, termed multiscale field of patterns (FoP). The paper discusses two example applications, contour integration and segmentation.

- Originality and Significance

The idea that multiscale representations can be useful to capture global image properties is at least as old as computer vision. Nonetheless, the paper presents a thorough and original derivation of a prior over higher-order binary features. In particular, if I understand correctly, the nonlinearity (logical OR) used to build the binary image pyramid, and the use of nonlinear filters (binary pattern detectors), allows the model to capture higher-order image properties. For instance, with the right choice of cost (or energy) functions for the binary patterns at different scales, the authors demonstrate that samples from the model trained on contour images can capture long and curved contours. This is an interesting result, and so are the results on contour detection in noisy images and figure/ground segmentation.

On the other hand, though, there are some important limitations. First, the authors only consider binary images and white pixel noise, therefore dealing with extremely simplified statistics, compared to natural images. For instance, given that the contour images are taken from the BSD (where both human-labeled contours and the original natural images are provided), it would be interesting to see if the model successfully detects contours also on those images (as opposed to just the binary contour image with white noise added on top). Neither do the authors experiment with white noise at different scales, which would be a slightly stronger test than adding white noise only at the finest level (since the noise averages out at coarser levels). Also, the authors compare the multiscale FoP model to a single scale FoP and to “baseline” models from the literature; but how well does the multiscale FoP compare to state of the art in the applications described?

Another major limitation, at present, seems the computational complexity of the algorithm (Line 300 and following: 2 days of training over a cluster with 20 cores for a 200 images training set?), which makes the approach rather impractical.

- Quality

The paper is technically sound and of good quality. It presents a simple and clear idea, develops the necessary math in great detail, and shows good results on interesting computer vision problems such as contour detection and figure/ground segmentation.

Minor: Line 418: “picked the pairwise discontinuity cost minimizing the per-pixel error rate in the test set”. Doesn’t this produce overfitting?

- Clarity

The paper is very well written and provides good coverage of relevant literature. I would suggest to consider also the work of Theis, L., Hosseini, R., & Bethge, M. (2012). Mixtures of conditional Gaussian scale mixtures applied to multiscale image representations. PLoS One, 7(7), e39857, which seems particularly relevant to this paper.

Typos:
Line 79: “fixed” is repeated twice.
Line 264: y_i should be y_1
Summary: A simple and clear model of high-order features in binary images based on nonlinear filtering at multiple scales. The model captures interesting image features and allows good performance in example applications; however the generalizability beyond binary images and white noise is not clear, and the computational complexity of learning seems a major obstacle.
Author Feedback
Author rebuttal: We thank the reviewers and area chair for their comments.

Below we address specific concerns from the reviewers, including the
question of how to apply the approach to realistic tasks.

Comments from Assigned_Reviewer_12:

"I have some concerns regarding the clarity
of presentation...in the learning part, it is not clear
if the parameters of the model are learned
from ground truth data, and if so, which
one? which parameters are trained, and
which are inferred from each image?"

The parameters that are learned are described in section 2.4. They
are the weights in the the log-linear representation of the MRF/CRF.
They parameterize the potential functions V^k and data model
functions D^k. We will clarify this in the paper.

Assigned_Reviewer_32, In the summary:

"however the generalizability
beyond binary images and white noise is
not clear, and the computational
complexity of learning seems a major
obstacle."

We have done experiments with more complex noise models but did
not include them because of space constraints. The results we
obtained were not very different from the ones with white noise. We
can include a brief description of these in the final version of the
paper.

The computational complexity of learning is significant but we don't
believe it is a major obstacle. In many applications learning is done
once and inference time is the main constraint. Our current
implementation for inference takes ~10 minutes on a large image from
the BSDS. We believe this could be improved using GPUs. We have
explored some possible GPU implementations and we plan to pursue that
further. Speeding up inference will also directly translate to a
speed up in learning time.

Comments from Assigned_Reviewer_4:

"But I am still left wondering what real world
task you would actually use this particular
model for."

On example is interactive segmentation. A foreground model can be
estimated from seeds or using EM as in Grab-cuts and similar methods.
In this case we would be replacing the Ising model with a FOP model
for the segmentation masks. With FOP we can have domain specific
models for masks. This is useful for example in medical applications.
The experimental results with leaf segmentation illustrate the
advantage of FOP over the Ising model for that task.

In some segmentation applications we have a graylevel or color
model for the pixels in the foreground and background specified in
advanced. In that case FOP can be used as a regularizer to estimate a
coherent segmentation.

Edge/boundary detection in natural images is another real world task.
It is possible for example to use FOP to "restore" the output of Pb or
another edge detection method into a clean binary contour map. One
challenge with the BSDS is that the ground truth contours are not
perfectly aligned with the boundaries in the input images. This
complicates learning but we are working on methods to address this.

"Also it would be nice if the authors could
show something about what the model has
learned. Is there somewhat to visualize w
and how it relates to the structure in
images?"

We can display the patterns with highest/lowest energy. We can
include this in the final version of the paper.